# Post-intensive care syndrome as a predictor of mortality in patients with critical illness: A cohort study

**Naoya Yanagi[1], Kentaro Kamiya[1,2]\*, Nobuaki Hamazaki[3], Ryota Matsuzawa[4], Kohei Nozaki[3], Takafumi Ichikawa[3], Thomas S. Valley[5,6,7], Takeshi Nakamura[1,8], Masashi Yamashita[1], Emi Maekawa[9], Tomotaka Koike[10], Minako Yamaoka-Tojo[2,11], Masayasu Arai[12], Atsuhiko Matsunaga[1,2], Junya Ako[9,11]**

1 Department of Rehabilitation Sciences, Graduate School of Medical Sciences, Kitasato University, Sagamihara, Japan, 2 Department of Rehabilitation, School of Allied Health Sciences, Kitasato University, Sagamihara, Japan, 3 Department of Rehabilitation, Kitasato University Hospital, Sagamihara, Japan, 4 Department of Physical Therapy, School of Rehabilitation, Hyogo University of Health Sciences, Kobe, Japan, 5 Division of Pulmonary and Critical Care Medicine, Department of Internal Medicine, University of Michigan, Ann Arbor, MI, United States of America, 6 Institute for Healthcare Policy and Innovation, University of Michigan, Ann Arbor, MI, United States of America, 7 Center for Bioethics and Social Sciences in Medicine, University of Michigan, Ann Arbor, MI, United States of America, 8 Department of Rehabilitation, Juntendo University Hospital, Tokyo, Japan, 9 Department of Cardiovascular Medicine, School of Medicine, Kitasato University, Sagamihara, Japan, 10 Department of Intensive Care Center, Kitasato University Hospital, Sagamihara, Japan, 11 Department of Cardiovascular Medicine, Graduate School of Medical Sciences, Kitasato University, Sagamihara, Japan, 12 Division of Intensive Care Medicine, Department of Research and Development Center for New Medical Frontiers, School of Medicine, Kitasato University, Sagamihara, Japan

\* k-kamiya@kitasato-u.ac.jp

**Data Availability Statement:** All relevant data are available in the Figshare repository at https://doi.org/10.6084/m9.figshare.14102567.v1.

## Abstract

### Introduction

The post-intensive care syndrome (PICS) encompasses multiple, diverse conditions, such as physical disability, cognitive impairment, and depression. We sought to evaluate whether conditions within PICS have similar associations with mortality among survivors of critical illness.

### Materials and methods

In this retrospective cohort study, we identified 248 critically ill patients with intensive care unit stay $\geq$72 hours, who underwent PICS evaluation. Patients with disability in activities of daily living, cognitive impairment, or depression before hospitalization were excluded. We defined PICS using established measures of physical disability (usual gait speed), cognitive impairment (Mini-Cog test), and depression (Patient Health Questionnaire-2) at hospital discharge. The endpoint was all-cause mortality.

### Results

Patients had a median age of 69 years and Acute Physiology and Chronic Health Evaluation (APACHE) II score of 16. One hundred thirty-two patients were classified as having PICS, and 19 patients died. 81/248 (34%) patients had physical disability, 42/248 (19%) had cognitive impairment, and 44/248 (23%) had depression. After adjusting for covariates on

**Funding:** The authors received no specific funding for this work.

**Competing interests:** The authors have declared that no competing interests exist.

multivariable Cox regression analyses, PICS was significantly associated with all-cause mortality (hazard ratio [HR] 3.78, 95% confidence interval [CI] 1.02 – 13.95; $P$ = 0.046). However, the association between PICS and all-cause mortality was related to physical disability and cognitive impairment ($P$ = 0.001 and $P$ = 0.027, respectively), while depression was not ($P$ = 0.623).

## Conclusion

While PICS as a syndrome has been useful in gaining attention to the sequelae of critical illness, its relationship with long-term mortality is driven largely by physical disability and cognitive impairment and not depression.

## Introduction

Mortality among critically ill patients in the intensive care unit (ICU) has decreased with the continued development of tools and techniques for providing life support [1]. Therefore, it is necessary to take long-term patient health and function into account when planning ICU care [2]. Post-intensive care syndrome (PICS) is a constellation of long-term physical, neurocognitive, and mental health complications in surviving patients following a stay in the ICU [3–7]. The term PICS describes new or worsening physical disability, cognitive impairment, or mental health impairment occurring after critical illness that persists after the period of hospitalization for acute care [8, 9].

Despite increasing awareness of its clinical importance, there have been few detailed studies regarding the evaluation of comprehensive PICS conditions in patients surviving after ICU treatment. In a population of 406 respiratory failure or shock patients in medical and surgical ICUs from five centers in the USA, Marra et al. reported that six of 10 patients surviving after a critical illness had one or more issues related to PICS up to 1 year after ICU admission [10]. In such cases, assessment of PICS conditions may be useful for risk stratifying survivors of critical illness. However, there is limited understanding as to whether PICS, as an umbrella syndrome, is associated with long-term mortality or whether any association is driven by a particular PICS condition.

## Materials and methods

### Setting and study participants

In this retrospective cohort study, 555 consecutive patients were identified who were admitted to the ICU of Kitasato University Hospital for more than 72 hours between April 2014 and February 2018 for cardiovascular disease. PICS evaluation was performed at hospital discharge. Patients who died during the hospitalization were excluded. Thirty-five patients were excluded because they were not independent in basic activities of daily living or had been diagnosed with dementia or depression before hospitalization (Fig 1). The study was performed in accordance with the tenets of the Declaration of Helsinki and was approved by the Ethics Committee of Kitasato University Hospital (B18-075).

### Data collection and outcome

Data on all variables were collected from electronic medical records. Baseline data, including age, sex, body mass index (BMI), and Charlson Comorbidity index (CCI) score [11], were

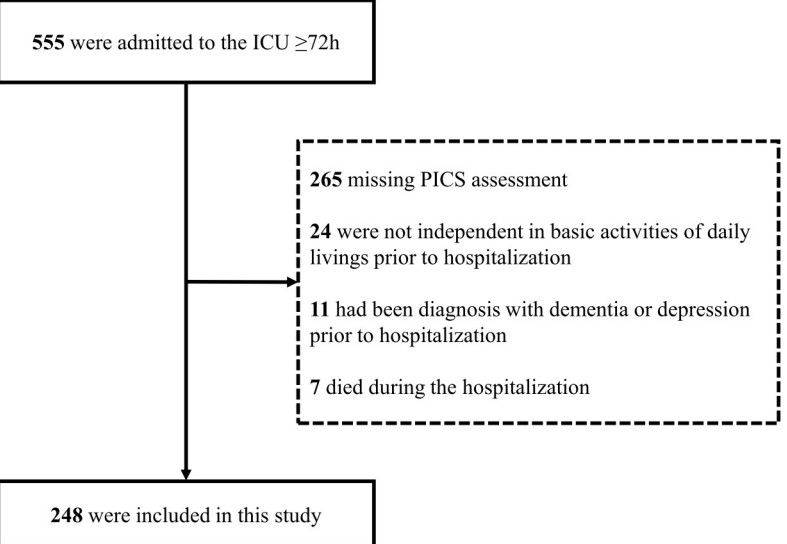

**Fig 1. Flow diagram of the patient selection and exclusion process in this study.**

collected. Clinical data, including clinical diagnosis (e.g., heart failure, acute coronary syndrome, cardiac surgery, aortic disease), Acute Physiology and Chronic Health Evaluation (APACHE) II score at admission [12], mean Sequential Organ Failure Assessment (SOFA) score during ICU stay [13], and the duration of hospital stay were also collected. In addition, the duration of mechanical ventilation, duration of delirium as assessed by the confusion assessment method for the intensive care unit (CAM-ICU) [14], the use of neuromuscular blocking agents, sedative drugs, or steroids and the requirement for intra-aortic balloon pumping, extra corporeal membrane oxygenation, or continuous veno-venous hemofiltration during the ICU stay, were recorded.

The primary outcome was the association between PICS and all-cause mortality. Secondary outcomes were the association between a PICS component (physical disability, cognitive impairment, or depression) and all-cause mortality. The endpoints were calculated as the number of days from discharge to the date of the event, and all-cause death was determined by a review of the medical records.

## Determining PICS components

We defined PICS using established measures of physical disability, cognitive impairment, and depression [15–17]. As a measure of physical disability, we used usual gait speed [18]. Usual gait speed was determined by measuring the speed at which the patient walked at their normal pace over a distance of 10 m. The use of walking aids was allowed for patients who usually used such equipment in their normal daily life. Physical disability was defined as a usual gait speed ≤0.8 m/s [19]. Patients who could not walk alone were categorized as physical disability. As a measure of cognitive function, we used the Mini-Cog test [20]. The Mini-Cog test consisted of a three-item recall test to assess memory and a clock-drawing test to assess executive function. The results of the clock-drawing test were assessed as normal (2 points) or abnormal (0 points), and Mini-Cog score ranged from 0 to 5. Cognitive impairment was defined as a Mini-Cog score ≤2 [21]. As a measure of depressive symptoms, we used the Patient Health Questionnaire-2 (PHQ-2) [22]. For the PHQ-2, the patients were asked how often they had experienced anhedonia and depressive symptoms in the past 2 weeks. Each question was given

a score from 0 to 3, and the total PHQ-2 score ranged from 0 to 6. Depression was defined as PHQ-2 score $\geq 3$ [23]. Evaluation of each PICS component was performed at the time of discharge from the hospital. A diagnosis of PICS was made in patients with at least one PICS component, while patients without a PICS component were defined as PICS-Free.

## Statistical analyses

Continuous variables were reported as the median and interquartile range (IQR), and categorical variables were expressed as number and percentage.

Baseline characteristics were compared by one-way analysis of variance, Wilcoxon's rank sum test, or the $\chi^2$ test as appropriate. The association between PICS and the primary outcome variable, all-cause mortality, was examined by the Kaplan–Meier method with the log-rank test and Cox proportional hazard model. We adjusted for the pre-existing risk factors including age, sex, BMI, CCI score, APACHE II score, and length of hospital stay, in the multivariable Cox proportional hazard model. We also examined the relationships between the different components of PICS (i.e., physical disability, cognitive impairment, and depression) and all-cause mortality using the Kaplan–Meier method with the log-rank test and multivariable Cox proportional hazard model adjusted for the pre-existing risk factors.

The multivariable logistic regression analysis was performed to predict risk factors for each component of PICS, using demographic covariates (age, sex, BMI and CCI score) and clinical covariates (APACHE II score, length of ICU stay, length of hospital stay, duration of mechanical ventilation, and duration of delirium).

Among patients who were reviewed, 16.9% had missing APACHE II data. We assumed that these data were missing at random and performed multiple imputation by chain equations with 20 iterations to replace missing values [24–26]. Age, sex, BMI, CCI score, APACHE II score, ICU length of stay, hospital length of stay, duration of mechanical ventilation and duration of delirium were incorporated into the imputation model. Estimates from each multiple data set were combined using Rubin's rule.

The analyses were performed using R (version 3.5.2; R Project for Statistical Computing, Vienna, Austria) and Stata, version 16.0. In all analyses, $P < 0.05$ was taken to indicate statistical significance.

## Results

### Patient characteristics

Fig 1 shows the patient flow in this study. Of the 555 patients who stayed in the ICU for 72 hours or more during the study period, 248 were included in the study. Overall, the participants had a median age of 69 years, 173/248 (70%) were male, and had a median APACHE II score at admission of 16. Diagnoses at admission were as follows: acute heart failure, 61/248 (24%); acute coronary syndrome, 64/248 (26%); admission following cardiac surgery, 32/248 (13%); aortic disease, 48/248 (19%); and other clinical diagnoses, 43/248 (17%). A diagnosis of PICS was made in 132/248 patients (53%) and a median hospital length of stay was 29 days (Table 1).

### Association of PICS with all-cause mortality

Following hospital discharge, 19 patients died over a mean follow-up period of 1.0 ± 0.8 years. Kaplan–Meier analysis indicated that patients with PICS had a significantly elevated mortality rate compared to patients without PICS (log-rank test: $P = 0.024$) (Fig 2). In the univariable and multivariable Cox regression analyses for all-cause mortality, PICS was also significantly

**Table 1. Patient characteristics.**

| Characteristics | All patients (n = 248) | PICS (n = 132) | PICS-Free (n = 116) | P value |
|---|---|---|---|---|
| Age (year), median (IQR) | 69 (58–77) | 74 (65–80) | 66 (54–71) | < 0.001 |
| Male sex, n (%) | 173 (70) | 87 (66) | 86 (74) | 0.168 |
| Body mass index (kg/m$^2$), median (IQR) | 22.1 (19.9–24.1) | 22.1 (19.7–24.1) | 22.1 (20.1–24.1) | 0.547 |
| Diagnoses at admission, n (%) | | | | |
| Acute heart failure | 61 (24) | 35 (27) | 26 (22) | 0.467 |
| Acute coronary syndrome | 64 (26) | 30 (23) | 34 (29) | 0.248 |
| Cardiac surgery | 32 (13) | 15 (11) | 17 (15) | 0.455 |
| Aortic disease | 48 (19) | 28 (21) | 20 (17) | 0.430 |
| Other diagnoses | 43 (17) | 23 (17) | 20 (17) | 0.970 |
| Charlson Comorbidity Index score, median (IQR) | 1 (0–2) | 1 (0–2) | 1 (0–2) | 0.798 |
| APACHE II score at admission, median (IQR) | 16 (12–22) | 17 (13–22) | 15 (11–19) | 0.005 |
| Mean daily Sequential Organ Failure Assessment score, median (IQR) | 6 (4–7) | 6 (4–7) | 6 (4–7) | 0.778 |
| Length of ICU stay, h, median (IQR) | 134 (98–195) | 158 (110–239) | 118 (92–161) | < 0.001 |
| Length of hospital stay, d, median (IQR) | 29 (21–43) | 35 (24–51) | 24 (19–34) | < 0.001 |
| Duration of mechanical ventilation, h, median (IQR) | 80 (0–137) | 88 (0–141) | 75 (0–124) | 0.144 |
| Delirium | | | | |
| Patients, n (%) | 144 (58) | 87 (66) | 57 (49) | 0.010 |
| Duration of delirium, d, median (IQR) | 1 (0–2) | 1 (0–3) | 0 (0–2) | < 0.001 |
| Neuromuscular blocking agents, n (%) | 17 (7) | 12 (9) | 5 (4) | 0.201 |
| Sedative, n (%) | 25 (10) | 20 (15) | 5 (4) | 0.005 |
| Steroid, n (%) | 76 (31) | 44 (33) | 32 (28) | 0.338 |
| Intra-aortic balloon pumping, n(%) | 29 (12) | 16 (12) | 13 (11) | 0.823 |
| Extra corporeal membrane oxygenation, n(%) | 7 (3) | 5 (4) | 2 (2) | 0.328 |
| Continuous veno-venous hemofiltration, n(%) | 11 (4) | 7 (5) | 4 (3) | 0.479 |
| Physical disability, n (%) | 81 (34) | 81 (66) | - | |
| Usual gait speed (m/s), median (IQR) | 0.99 (0.76–1.18) | 0.74 (0.55–0.95) | 1.12 (0.99–1.26) | < 0.001 |
| Cognitive impairment, n (%) | 42 (19) | 42 (42) | - | |
| Mini-Cog score, median (IQR) | 5 (3–5) | 3 (2–5) | 5 (4–5) | < 0.001 |
| Depression, n (%) | 44 (23) | 44 (56) | - | |
| PHQ-2 score, median (IQR) | 1 (0–2) | 3 (1–4) | 0 (0–2) | < 0.001 |
| Death, n (%) | 19 (8) | 16 (12) | 3 (0) | 0.007 |

Values are the median (IQR) or % of total.

IQR = interquartile range; APACHE II = Acute Physiology and Chronic Health Evaluation II; ICU = intensive care unit; PHQ = Patient Health Questionnaire.

associated with all-cause mortality after adjusting for covariates (unadjusted: hazard ratio [HR] 3.79, 95% confidence interval [CI] 1.10 – 13.12; *P* = 0.035; adjusted: HR 3.78, 95% CI 1.02 – 13.95; *P* = 0.046).

With regard to the associations between PICS components and mortality, physical disability and cognitive impairment at discharge were shown to be significantly associated with higher rate of all-cause mortality on Kaplan–Meier analysis (log-rank test: *P* = 0.001 and *P* = 0.027, respectively). In contrast, depression at discharge was not significantly associated with mortality (log-rank test: *P* = 0.623) (Fig 3). In the multivariable Cox proportional hazard model, physical disability was significantly associated with all-cause mortality after adjusting for covariates (HR 4.19, 95% CI 1.39–12.59; *P* = 0.011), but cognitive impairment and depression were not significantly associated with all-cause mortality (HR 2.46, 95% CI 0.77–7.84; *P* = 0.129 and HR 1.28, 95% CI 0.30–5.65; *P* = 0.743).

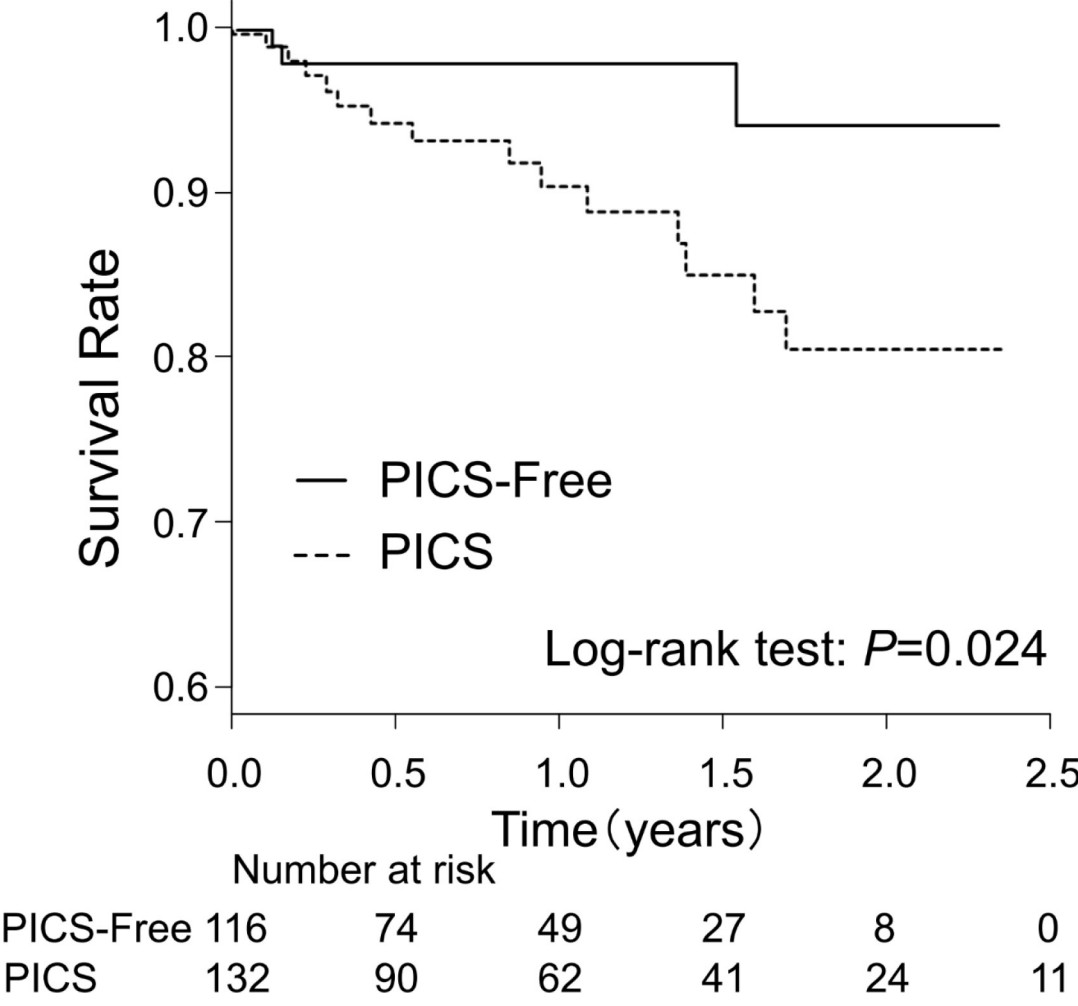

**Fig 2. Kaplan–Meier analysis of all-cause mortality according to Post-Intensive Care Syndrome (PICS) components.** PICS group, patients with at least one PICS component; PICS-Free group, patients with no PICS components.

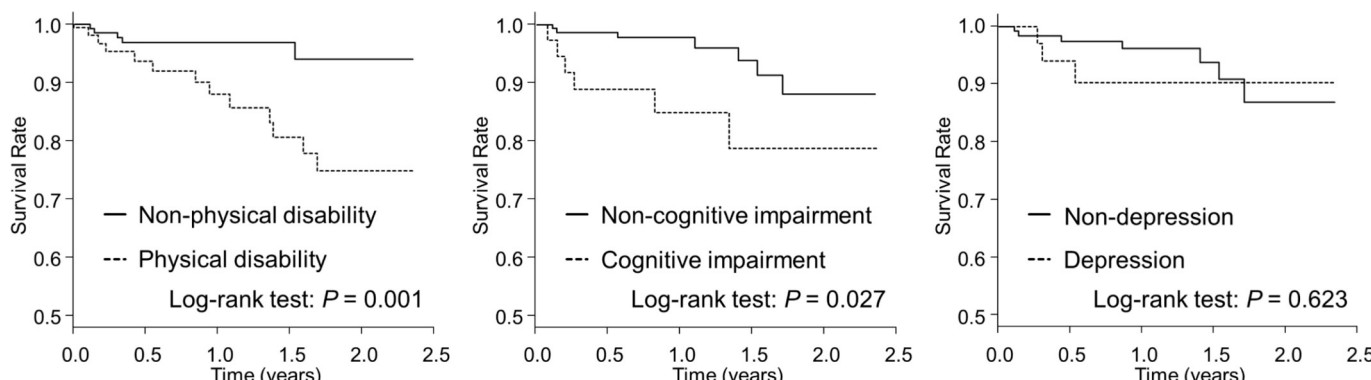

**Fig 3. Kaplan–Meier analysis of all-cause mortality according to physical disability, cognitive impairment, and depression.** Physical disability was defined as usual gait speed ≤0.8 m/s, cognitive impairment by Mini-Cog score ≤2, and depression according to Patient Health Questionnaire-2 (PHQ-2) score ≥3.

**Table 2. Association of clinical variables and conditions with each component of post-intensive care syndrome.**

| Variables | Physical disability | | | Cognitive impairment | | | Depression | | |
| --- | --- | --- | --- | --- | --- | --- | --- | --- | --- |
| | Usual gait speed $\leq$ 0.8 m/s | | | Mini-Cog score $\leq$ 2 | | | PHQ-2 score $\geq$ 3 | | |
| | OR | 95% CI | *P* value | OR | 95% CI | *P* value | OR | 95% CI | *P* value |
| Age | 1.08 | 1.05–1.12 | < 0.001 | 1.07 | 1.02–1.11 | 0.004 | 1.03 | 1.00–1.06 | 0.048 |
| Male | 0.82 | 0.42–1.61 | 0.573 | 0.95 | 0.41–2.18 | 0.903 | 1.64 | 0.70–3.83 | 0.258 |
| BMI (per 1.0 kg/m$^2$) | 1.05 | 0.97–1.14 | 0.244 | 0.93 | 0.83–1.05 | 0.251 | 0.92 | 0.82–1.03 | 0.157 |
| CCI score | 1.01 | 0.86–1.19 | 0.898 | 0.98 | 0.79–1.21 | 0.826 | 1.11 | 0.93–1.35 | 0.235 |
| APACHE II score | 0.99 | 0.95–1.04 | 0.938 | 1.06 | 1.00–1.12 | 0.040 | 0.99 | 0.94–1.05 | 0.765 |
| Length of ICU stay (per 1 day) | 1.13 | 1.04–1.22 | 0.006 | 1.03 | 0.94–1.12 | 0.518 | 0.96 | 0.85–1.07 | 0.447 |
| Length of hospital stay (per 1 day) | 1.03 | 1.01–1.04 | 0.004 | 0.97 | 0.94–0.99 | 0.022 | 1.02 | 0.99–1.04 | 0.143 |
| Duration of mechanical ventilation (per 1 day) | 0.97 | 0.86–1.09 | 0.576 | 1.01 | 0.88–1.17 | 0.889 | 1.13 | 0.99–1.29 | 0.056 |
| Duration of delirium (per 1 day) | 0.91 | 0.77–1.07 | 0.264 | 1.45 | 1.18–1.77 | < 0.001 | 0.95 | 0.80–1.14 | 0.606 |

OR = odds ratio. CI = confidence interval. BMI = body mass index, CCI = Charlson Comorbidity index, APACHE II = Acute Physiology and Chronic Health Evaluation II, ICU = Intensive Care Unit.

### Predictors of each PICS component

Table 2 shows the results of multivariable logistic regression analysis to predict the occurrence of each PICS components. After adjusting for covariates, increasing age (odds ratio [OR] 1.08, 95% CI 1.05 – 1.12; *P* < 0.001), longer ICU stay (OR 1.13, 95% CI 1.04 – 1.22; *P* = 0.006) and longer duration of hospital stay (OR 1.03, 95% CI 1.01–1.04; *P* = 0.004) were independent predictors of the occurrence of physical disability. For cognitive impairment, increasing age (OR 1.07, 95% CI 1.02 – 1.11; *P* = 0.004), greater severity of illness (OR 1.06, 95% CI 1.00 – 1.12; *P* = 0.040), longer duration of hospital stay (OR 0.97, 95% CI 0.94–0.99; *P* = 0.022), and for each day of prolonged delirium (OR 1.45, 95% CI 1.18 – 1.77; *P* < 0.001) were independent predictors. For depression, increasing age (OR 1.03, 95% CI 1.00 – 1.06; *P* = 0.048) was an independent predictor.

## Discussion

This study investigated whether conditions within PICS had similar associations with all-cause mortality in survivors of critical illness. Results demonstrated that having PICS was independently associated with long-term mortality. Among the conditions with PICS, physical disability and cognitive impairment were associated with long-term mortality but not depression. In addition, our results suggest that increasing age is an important independent predictor for each PICS condition.

Physical and mental dysfunction, components of PICS, have been reported to be associated with poor outcomes in critically ill patients [5, 6, 27]. One study showed that PICS, defined by physical disability, cognitive impairment, and depression, occurred at a rate of 56% at 12 months after leaving the ICU [10]. New and worsening physical and mental dysfunction after critical illness were reported to be associated with increased in-hospital and 1-year mortality rates [3–5, 28–30]. To our knowledge, however, there have been no reports regarding the relationship between systematic PICS evaluation and mortality among ICU survivors. In the present study, PICS was associated with a nearly 4-fold increase in mortality risk after adjusting for severity of illness. In addition, physical disability and cognitive impairment following critical illness were also associated with long-term mortality.

Physical disability, as defined by slow gait speed, was shown to be independently associated with increased risk of mortality. Due to population aging, the integration of sarcopenia or

frailty as a vital sign in older patients is becoming increasingly important as a means of guiding management and coordinating clinical care [31]. Sarcopenia/frailty is identified by measuring gait speed [19, 32], which is a suitable functional test for pharmacological trials in older adults. Our recent study in a population of 1474 older patients with cardiovascular disease showed that the prognostic capability of gait speed measurement was comparable to that of the six-minute walking distance [17]. In addition, short-term improvement of gait speed in older patients with acute heart failure while in hospital was significantly associated with reduced risks of death and readmission [33]. Gait speed was reported to be a reliable and valid measure of physical function in acute respiratory distress syndrome patients who survived after hospitalization in the ICU [18]. Changes in gait speed were also reported to be consistent with changes in physical function, and the estimated minimal important difference of gait speed was 0.03 – 0.06 m/s. Gait speed is determined by skeletal muscle mass and muscle strength, and higher values of both were reported to have protective effects against disease [34–40]. In addition, both muscle mass and strength have been shown to be strongly correlated with exercise capacity [17], nutritional status [41], and inflammatory marker levels [42, 43]. Sharshar et al. suggested that the association of muscle weakness with high mortality risk may be explained by increased risk of infections [4]. Taken together, these previous reports and the findings of the present study suggest that physical disability and muscle dysfunction are critically related to prognosis, and support the use of gait speed for assessment of physical disability in critically ill patients and for long-term prognostication at the time of discharge.

Approximately 30% to 80% of ICU survivors show cognitive impairment, which often lasts for many years [6]. Psychiatric conditions, such as anxiety, depression, and post-traumatic stress disorder, have been reported to occur in 7% – 59% of patients, and these may also persist for several years [27, 30, 44, 45]. In our study, cognitive impairment, but not depression, was associated with elevated mortality risk. A number of major risk factors and possibly underlying biological mechanisms for post-ICU cognitive dysfunction have been reported previously. For example, older age [46] and delirium [47] were shown to be the major risk factors associated with the occurrence of long-term cognitive impairment in ICU survivors. Hypoglycemia, hyperglycemia, hypoxaemia, hypotension, and sedation may also be associated with long-term cognitive impairment in such cases [44, 46]. These associations may explain the observed relation between cognitive impairment and prognosis in critically ill patients, and reiterate the importance of developing methods to identify cognitive impairment in ICU survivors.

We identified several potential risk factors for each PICS condition. Increasing age, length of ICU stay, and length of hospital stay were predictors of physical disability; increasing age, severity of illness, length of hospital stay, and duration of delirium were predictors of cognitive impairment; and increasing age was a predictor of depression. Among these factors, an extended length of ICU or hospital stay and delirium prevention may be key targets for reducing the incidence of PICS. Recent clinical trials over the last 10 years have reported that pharmacotherapy and treatment bundles including early ambulation showed positive effects in shortening the length of ICU stay and delirium [48–50]. Furthermore, patients of increasing age may be targeted to enrich future PICS trials or for preferential referral to post-ICU clinics, which may be helpful to ICU survivors.

This study should be considered in the context of several limitations. First, this was a single-center retrospective study. At the same time, we present a large study population with characteristics that are largely generalizable to diverse settings. Second, only patients with cardiovascular disease were included in the present study; however, the existing literature does not suggest that specific critical illness conditions might have different outcomes related to PICS. Third, in the present study, the PHQ-2 was used to assess the mental health component of PICS, which has been shown to be useful as a screening tool for depression [22], but as a

diagnostic tool, further validation is necessary. In addition, we did not assess other mental health measures, such as anxiety and post-traumatic stress disorder, in the present study. Further research on mental health measures is needed. Fourth, since we only assessed delirium during the ICU stay, the impact of delirium persistent after ICU discharge on PICS is unclear. Finally, we did not collect data on other potential confounders, such as preadmission physical activity, exercise capacity, or socioeconomic status.

## Conclusions

While PICS as a syndrome has been useful in gaining attention to the sequelae of critical illness, its relationship with long-term mortality is driven largely by physical disability and cognitive impairment and not depression. This heterogeneity suggests that future trials to improve care for patients with PICS should target specific PICS conditions, rather than the PICS syndrome itself.

## Acknowledgments

We would like to thank the patients and the research team for their participation in this study.

## Author Contributions

**Conceptualization:** Naoya Yanagi.

**Data curation:** Naoya Yanagi, Nobuaki Hamazaki, Ryota Matsuzawa, Kohei Nozaki, Takafumi Ichikawa, Takeshi Nakamura, Masashi Yamashita, Emi Maekawa.

**Formal analysis:** Naoya Yanagi, Kentaro Kamiya, Thomas S. Valley, Takeshi Nakamura.

**Methodology:** Naoya Yanagi, Kentaro Kamiya, Thomas S. Valley, Tomotaka Koike.

**Project administration:** Thomas S. Valley.

**Supervision:** Kentaro Kamiya, Atsuhiko Matsunaga.

**Writing – original draft:** Naoya Yanagi, Kentaro Kamiya, Thomas S. Valley.

**Writing – review & editing:** Kentaro Kamiya, Nobuaki Hamazaki, Ryota Matsuzawa, Kohei Nozaki, Takafumi Ichikawa, Thomas S. Valley, Takeshi Nakamura, Masashi Yamashita, Emi Maekawa, Tomotaka Koike, Minako Yamaoka-Tojo, Masayasu Arai, Atsuhiko Matsunaga, Junya Ako.

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
