## [Decision Letter · Decision Letter 0]

6 Oct 2020

PONE-D-20-29849

Post-Intensive Care Syndrome as a Predictor of Mortality in Patients with Critical Illness: a cohort study

PLOS ONE

Dear Dr. Kamiya,

Thank you for submitting your manuscript to PLOS ONE. After careful consideration, we feel that it has merit but does not fully meet PLOS ONE’s publication criteria as it currently stands. Therefore, we invite you to submit a revised version of the manuscript that addresses the points raised during the review process.

Please revise accordingly.

We look forward to receiving your revised manuscript.

Kind regards,

Academic Editor

PLOS ONE

Journal Requirements:

Reviewers' comments:

Reviewer's Responses to Questions

**Comments to the Author**

1. Is the manuscript technically sound, and do the data support the conclusions?

Reviewer #1: Yes

Reviewer #2: Yes

2. Has the statistical analysis been performed appropriately and rigorously? 

Reviewer #1: Yes

Reviewer #2: Yes

3. Have the authors made all data underlying the findings in their manuscript fully available?

Reviewer #1: Yes

Reviewer #2: Yes

4. Is the manuscript presented in an intelligible fashion and written in standard English?

Reviewer #1: Yes

Reviewer #2: Yes

5. Review Comments to the Author

Reviewer #1: The authors conducted a retrospective cohort study to evaluate whether conditions within post-intensive care syndrome (PICS) have similar associations with mortality among 248 survivors of critical illness. They found that 132 (53%) of the patients had PICS and 19 (7.6%) died during follow-up. PICS was significantly associated with mortality after adjusting for age and APACHE II score. The association between PICS and mortality was related to physical disability and cognitive impairment, but not depression. They concluded that while PICS as a syndrome has been useful in gaining attention to the sequelae of critical illness, its relationship with long-term mortality is driven largely by physical disability and cognitive impairment and not depression. The study is of interest and provides novel findings on the association between PICS and long-term mortality among ICU survivors. There are no major issues in the study design and analyses. Some minor issues are given below.

1. How was delirium assessed and defined? Was it assessed only in the ICU or also in the ward? If delirium persisted at hospital discharge, how was PICS evaluated?

2. The duration of follow-up after hospital discharge was not defined.

3. Only age and APACHE II score were adjusted in the model for mortality. Why were sex, BMI, comorbidity, delirium and hospital length of stay (LOS) were not included as confounders for mortality?

4. Similarly, why hospital LOS was not included as a confounder in modelling the risk factors for each component of PICS?

5. In the results, the authors mentioned that 35 patients were excluded because they were not independent in basic activities of daily livings or had been diagnosed with dementia or depression prior to hospitalization. These exclusion criteria should be stated in the methods, instead of the results.

6. Please specify the period that the “Mean daily Sequential Organ Failure Assessment score” were calculated. Did the period include all ICU stay or only a few days?

7. Please provide references for the statement in Page 16, lines 258-259, and check the grammer (“In recent clinical trials for 10 years, it has been ….”)

8. The components of psychiatric disability included depression, anxiety and post-traumatic stress disorder. However, only depression was measured by a screening tool in the study. Moreover, according to a validation study, using an independent structured mental health professional re-interview as the criterion standard, a Patient Health Questionnaire-2 (PHQ-2) score ≥3 had a sensitivity of 83% and a specificity of 92% for major depression. The PHQ-2 score helps to screen, not to diagnose, patients with depression. This should be addressed in the study limitation.

Reviewer #2: This study investigated the association between Post-Intensive Care Syndrome (PICS) and the outcome of critically ill patients. The authors found that PICS was significantly associated with all-cause mortality (hazard ratio [HR] 4.12, 95% CI, 1.15 – 14.85; P = 0.030). Moreover, the association between PICS and all-cause mortality was related to physical disability and cognitive impairment (P = 0.001 and P = 0.027, respectively), while depression was not (P =

0.623). Overall, the study was well-designed and the manuscript was well-written. I just have several suggestions.

1. The manuscript should be improved after English editing by a native speaker.

2. Please define the timing of primary outcome - all cause mortality.

3. Please add more detail ICU admission diagnosis in table 1. In the present form, 37% was classifed as other cause.

4. Please make a figure to show the study algorithm and briefly describe the characteristics of ICUs in this study and what kind of patients would undergo PICS evaluation and its associated percentage.

5. Pleasse add the interventions, including IABP, ECMO and CVVH.

6. Please show what variable you adjusted in the analysis of the assciation between PICS and mortality and also explain the rationale.

6. PLOS authors have the option to publish the peer review history of their article (what does this mean?). If published, this will include your full peer review and any attached files.

Reviewer #1: **Yes: **Hsiu-Nien Shen

Reviewer #2: No

---

## [Author Response · Author response to Decision Letter 0]

10 Dec 2020

Dear Reviewers and Editors,

Thank you for reviewing our manuscript and for your valuable advice. In this letter, the reviewers’ comments are presented in bold, followed by our point-by-point responses.

Comments from Reviewer 1

Comment 1. How was delirium assessed and defined? Was it assessed only in the ICU or also in the ward? If delirium persisted at hospital discharge, how was PICS evaluated? 

Response: We evaluated delirium using the confusion assessment method for the intensive care unit (CAM-ICU). Delirium was only assessed in the ICU. We have changed the text that described delirium assessment in the Methods section as follows: 

Page 6, lines 17-19 (Methods section)

From

“In addition, the duration of mechanical ventilation, duration of delirium (14), …”

To

“In addition, the duration of mechanical ventilation, duration of delirium as assessed by the confusion assessment method for the intensive care unit (CAM-ICU) (14), …”

Unfortunately, we did not have data on delirium at the time when we performed the PICS evaluation. However, the patients did not have any problems understanding the instructions for the PICS evaluation. To clarify this point, we have added additional details to the Discussion section as follows:

Page 19, lines 10-12 (Discussion section)

“Fourth, since we only assessed delirium during the ICU stay, the impact of persistent delirium after ICU discharge on PICS is unclear.”

Comment 2. The duration of follow-up after hospital discharge was not defined.

Response: In the original manuscript, the mean follow-up period was noted in the Results section as follows: 

Page 13, lines 2-3 (Results section)

“Following hospital discharge, 19 patients died over a mean follow-up period of 1.0 ± 0.8 years.”

Comment 3. Only age and APACHE II score were adjusted in the model for mortality. Why were sex, BMI, comorbidity, delirium and hospital length of stay (LOS) were not included as confounders for mortality?

Response: Thank you for pointing this out. We added the factors that you have suggested to the multivariate analysis, and the results were similar to the original outcomes. In addition, we revised the text as follows: 

Page 8, lines 14-19 (Methods section)

From

“We adjusted for age and APACHE II score in the multivariable Cox proportional hazard model. We also examined the relationships between the components of PICS (i.e., physical disability, cognitive impairment, and depression) and all-cause mortality using the Kaplan–Meier method with the log-rank test and multivariable Cox proportional hazard model adjusted for age and APACHE II score.”

To

“We adjusted for the pre-existing risk factors including age, sex, BMI, CCI score, APACHE II score, and length of hospital stay, in the multivariable Cox proportional hazard model. We also examined the relationships between the different components of PICS (i.e., physical disability, cognitive impairment, and depression) and all-cause mortality, using the Kaplan–Meier method with the log-rank test and multivariable Cox proportional hazard model adjusted for the pre-existing risk factors.”

Page 13, lines 4-7 (Results section)

From

“In univariable and multivariable Cox regression analyses for all-cause mortality, PICS was also significantly associated with all-cause mortality after adjusting for age and APACHE II score (unadjusted: hazard ratio [HR] 3.79, 95% confidence interval [CI] 1.10 – 13.12; P = 0.035; adjusted: HR 4.12, 95% CI 1.15 – 14.85; P = 0.030).”

To

“In the univariable and multivariable Cox regression analyses for all-cause mortality, PICS was also significantly associated with all-cause mortality after adjusting for covariates (unadjusted: hazard ratio [HR] 3.79, 95% confidence interval [CI] 1.10 – 13.12; P = 0.035; adjusted: HR 3.78, 95% CI 1.02 – 13.95; P = 0.046).”

Page 13, lines 18-19 and Page 14, lines 1-3 (Results section)

From

“In multivariable Cox proportional hazard model, physical disability was significantly associated with all-cause mortality after adjusted for age and APACHE II score (HR 4.39, 95% CI 1.57 – 12.27; P = 0.005), but cognitive impairment and depression were not significantly associated with all-cause mortality (HR 2.46, 95% CI 0.77 – 7.89; P = 0.129 and HR 1.37, 95% CI 0.26 – 7.28; P = 0.709).”

To

“In the multivariable Cox proportional hazard model, physical disability was significantly associated with all-cause mortality after adjusting for covariates (HR 4.19, 95% CI 1.39 – 12.59; P = 0.011), but cognitive impairment and depression were not significantly associated with all-cause mortality (HR 2.46, 95% CI 0.77 – 7.84; P = 0.129 and HR 1.28, 95% CI 0.30 – 5.65; P = 0.743).”

Comment 4. Similarly, why hospital LOS was not included as a confounder in modelling the risk factors for each component of PICS?

Response: Thank you for highlighting this point. As suggested, we added hospital LOS as a confounding factor and performed the analysis again. As a result, hospital LOS was identified as a significant factor for disability. The results of this analysis have been incorporated into the revised manuscript as follows:

Page 9, lines 1-4 (Methods section)

From

“Multivariable logistic regression analysis was performed to predict risk factors for each component of PICS, using demographic covariates (age, sex, BMI and CCI score) and clinical covariates (APACHE II score, ICU length of stay, duration of mechanical ventilation and duration of delirium).”

To

“The multivariate logistic regression analysis was performed to predict risk factors for each component of PICS using demographic covariates (age, sex, BMI, and CCI score) and clinical covariates (APACHE II score, length of ICU stay, length of hospital stay, duration of mechanical ventilation, and duration of delirium).”

Page 14, lines 12-19 (Results section)

From

“After adjusting for covariates, increasing age (odds ratio [OR] 1.08, 95% CI 1.05 – 1.11; P < 0.001) and longer ICU stay (OR 1.18, 95% CI 1.09 – 1.28; P < 0.001) were independent predictors of the occurrence of physical disability. For cognitive impairment, increasing age (OR 1.06, 95% CI 1.02 – 1.10; P = 0.003), higher severity of illness (OR 1.07, 95% CI 1.01 – 1.12; P = 0.018) and for each day of prolonged delirium (OR 1.33, 95% CI 1.06 – 1.66; P = 0.013) were independent predictors. For depression, increasing age (OR 1.03, 95% CI 1.00 – 1.07; P = 0.047) and lower BMI (OR 0.99, 95% CI 0.98 – 0.99; P = 0.044) were independent predictors.”

To

“After adjusting for covariates, increasing age (odds ratio [OR] 1.08, 95% CI 1.05 – 1.12; P < 0.001), longer ICU stay (OR 1.13, 95% CI 1.04 – 1.22; P = 0.006) and longer duration of hospital stay (OR 1.03, 95% CI 1.01 – 1.04; P = 0.004) were independent predictors of the occurrence of physical disability. For cognitive impairment, increasing age (OR 1.07, 95% CI 1.02 – 1.11; P = 0.004), greater severity of illness (OR 1.06, 95% CI 1.00 – 1.12; P = 0.040), longer duration of hospital stay (OR 0.97, 95% CI 0.94 – 0.99; P = 0.022), and for each day of prolonged delirium (OR 1.45, 95% CI 1.18 – 1.77; P < 0.001) were independent predictors. For depression, increasing age (OR 1.03, 95% CI 1.00 - 1.06; P = 0.048) was an independent predictor.”

Page 18, lines 12-16 (Discussion section)

From

“We identified several potential risk factors for each PICS condition. Increasing age and ICU length of stay were predictors of physical disability; increasing age, high severity, and duration of delirium were predictors of cognitive impairment; and increasing age and low BMI were predictors of depression. Among these factors, extended ICU length of stay and delirium prevention may be key targets to reducing the incidence of PICS.”

To

“We identified several potential risk factors for each PICS condition. Increasing age, length of ICU stay, and length of hospital stay were predictors of physical disability; increasing age, severity of illness, length of hospital stay, and duration of delirium, were predictors of cognitive impairment; and increasing age was a predictor of depression. Among these factors, an extended length of ICU or hospital stay and delirium prevention may be key targets for reducing the incidence of PICS.”

Comment 5. In the results, the authors mentioned that 35 patients were excluded because they were not independent in basic activities of daily livings or had been diagnosed with dementia or depression prior to hospitalization. These exclusion criteria should be stated in the methods, instead of the results.

Response: As suggested, the sentence on exclusion criteria has been moved from the Results section to the Methods section as follows: 

Page 6, lines 3-5 (Methods section)

“Thirty-five patients were excluded because they were not independent in basic activities of daily living or had been diagnosed with dementia or depression before hospitalization (Fig 1).”

Comment 6. Please specify the period that the “Mean daily Sequential Organ Failure Assessment score” were calculated. Did the period include all ICU stay or only a few days?

Response: We calculated the “mean daily Sequential Organ Failure Assessment score” by averaging all of the SOFA scores over the course of their ICU stay. We corrected the text as follows:

Page 6, lines 16-17 (Methods section)

From

“… (12), mean Sequential Organ Failure Assessment (SOFA) score (13) and hospital length of stay were also collected.”

To

“… (12), mean Sequential Organ Failure Assessment (SOFA) score during ICU stay (13), and the duration of hospital stay were also collected.”

Comment 7. Please provide references for the statement in Page 16, lines 258-259, and check the grammar (“In recent clinical trials for 10 years, it has been ….”)

Response: As suggested, we have modified the description and added the relevant references in the Discussion as follows: 

Page 18, lines 16-19 (Discussion section)

From

“In recent clinical trials for 10 years, it has been reported that pharmacotherapy and treatment bundles including early ambulation have the effect in shortening ICU duration and delirium.”

To

“Recent clinical trials over the last 10 years have reported that pharmacotherapy and treatment bundles including early ambulation showed positive effects in shortening the length of ICU stay and delirium (48-50).”

[Reference]

48. Kayambu G, Boots R, Paratz J. Physical therapy for the critically ill in the ICU: a systematic review and meta-analysis. Crit Care Med. 2013;41(6):1543-54. Epub 2013/03/27. doi: 10.1097/CCM.0b013e31827ca637. PubMed PMID: 23528802.

49. Schweickert WD, Pohlman MC, Pohlman AS, Nigos C, Pawlik AJ, Esbrook CL, et al. Early physical and occupational therapy in mechanically ventilated, critically ill patients: A randomized controlled trial. The Lancet. 2009;373(9678):1874-82. doi: 10.1016/s0140-6736(09)60658-9.

50. Needham DM, Korupolu R, Zanni JM, Pradhan P, Colantuoni E, Palmer JB, et al. Early physical medicine and rehabilitation for patients with acute respiratory failure: a quality improvement project. Arch Phys Med Rehabil. 2010;91(4):536-42. Epub 2010/04/13. doi: 10.1016/j.apmr.2010.01.002. PubMed PMID: 20382284.

Comment 8. The components of psychiatric disability included depression, anxiety and post-traumatic stress disorder. However, only depression was measured by a screening tool in the study. Moreover, according to a validation study, using an independent structured mental health professional re-interview as the criterion standard, a Patient Health Questionnaire-2 (PHQ-2) score ≥3 had a sensitivity of 83% and a specificity of 92% for major depression. The PHQ-2 score helps to screen, not to diagnose, patients with depression. This should be addressed in the study limitation.

Response: Thank you for your suggestion. We have made the necessary changes to the Discussion section as follows:

Page 19, lines 6-10 (Discussion section)

“Third, in the present study, the PHQ-2 was used to assess the mental health component of PICS, which has been shown to be useful as a screening tool for depression (22), but as a diagnostic tool, further validation is necessary. In addition, we did not assess other mental health measures, such as anxiety and post-traumatic stress disorder, in the present study. Further research on mental health measures is needed.”

 

Comments from Reviewer 2

Comment 1. The manuscript should be improved after English editing by a native speaker.

Response: As suggested, the manuscript has been proofread by a native English speaker.

Comment 2. Please define the timing of primary outcome - all cause mortality.

Response: In the original manuscript, all-cause mortality was determined by reviewing medical records. We have stated this in the Methods section as follows:

Page 7, lines 5-7 (Methods section)

“The endpoints were calculated as the number of days from discharge to the date of the event, and all-cause death was determined by a review of the medical records.”

Comment 3. Please add more detail ICU admission diagnosis in table 1. In the present form, 37% was classified as other cause. 

Response: As suggested, we have modified the Results and Table 1 as follows:

Page 10, line 1-3 (Results section)

From

“Diagnoses at admission were acute heart failure: 61/248 (24%), acute coronary syndrome: 64/248 (26%), cardiac surgery: 32/248 (13%), or other clinical diagnoses, 91/248 (37%).” 

To

“Diagnoses at admission were as follows: acute heart failure, 61/248 (24%); acute coronary syndrome, 64/248 (26%); admission following cardiac surgery, 32/248 (13%); aortic disease, 48/248 (19%); and other clinical diagnoses, 43/248 (17%).”

Comment 4. Please make a figure to show the study algorithm and briefly describe the characteristics of ICUs in this study and what kind of patients would undergo PICS evaluation and its associated percentage.

Response: As suggested, we have included the patient flow chart. In this study, PICS assessment was performed on approximately half of all patients who stayed in the ICU for ≥ 72 h. We have modified the Methods and Results as follows:

Page 5, lines 19 and Page 6, lines 1-2 (Methods section)

From

“In this retrospective cohort study, 283 consecutive patients were identified who underwent PICS evaluation and were admitted to the ICU of Kitasato University Hospital for more than 72 hours between April 2014 and February 2018 for cardiovascular disease.”

To

“In this retrospective cohort study, 555 consecutive patients were identified who were admitted to the ICU of Kitasato University Hospital for more than 72 hours between April 2014 and February 2018 for cardiovascular disease.”

Page 9, lines 17-18 (Results section)

From

“This study was performed in a total of 248 patients.”

To

“Figure 1 shows the patient flow in this study. Of the 555 patients who stayed in the ICU for 72 hours or more during the study period, 248 were included in the study.” 

Fig 1. Flow diagram of the patient selection and exclusion process in this study.

Comment 5. Please add the interventions, including IABP, ECMO and CVVH.

Response: As suggested, we have modified the Methods section as follows: 

Page 6, lines 19 and Page 7, lines 1-2 (Methods section)

From

“… and the use of neuromuscular blocking agents, sedative drugs, or steroids during the ICU stay were recorded.”

To

“… the use of neuromuscular blocking agents, sedative drugs, or steroids and the requirement for intra-aortic balloon pumping, extracorporeal membrane oxygenation, or continuous veno-venous hemofiltration during the ICU stay, were recorded.”

Comment 6. Please show what variable you adjusted in the analysis of the association between PICS and mortality and also explain the rationale.

Response: We adjusted for pre-existing risk factors in the analysis of the association between PICS and mortality. (Please also see the response to Comment 3 from Reviewer 1 about the adjustment variables we used.) The number of events was small; therefore, we preferentially adjusted for the main factors used in previous cohort studies [1-4]. 

Reference

1. Hatch R, Young D, Barber V, Griffiths J, Harrison DA, Watkinson P. Anxiety, Depression and Post traumatic Stress Disorder after critical illness: a UK-wide prospective cohort study. Crit Care. 2018;22(1):310. Epub 2018/11/24 doi: 10.1186/s13054-018-2223-6. PubMed PMID: 30466485. PubMed Central PMCID: PMC6251214.

2. Brummel NE, Bell SP, Girard TD, Pandharipande PP, Jackson JC, Morandi A, et al. Frailty and subsequent disability and mortality among patients with critical illness. Am J Respir Crit Care Med. 2017;196(1):64. Epub 2016/12/07 doi: 10.1164/rccm.201605-0939OC. PubMed PMID: 27922747. PubMed Central PMCID: PMC5519959.

3. Basile-Filho A, Lago AF, Menegueti MG, Nicolini EA, Rodrigues LAB, Nunes RS, et al. The use of Apache II, SOFA, SAPS 3, C-reactive protein/albumin ratio, and lactate to predict mortality of surgical critically ill patients: A retrospective cohort study. Med (Baltim). 2019;98(26):e16204. Epub 2019/07/03 doi: 10.1097/MD.0000000000016204. PubMed PMID: 31261567. PubMed Central PMCID: PMC6617482.

4. Bagshaw SM, Stelfox HT, McDermid RC, Rolfson DB, Tsuyuki RT, Baig N, et al. Association between frailty and short- and long-term outcomes among critically ill patients: a multicentre prospective cohort study. C.M.A.J Can Med Assoc J. 2014;186(2):E95. doi: 10.1503/cmaj.130639. PubMed PMID: PMID: WOS: 000330267300016.

We declare that all modifications are listed in the file with “with TRACK CHANGES” and do not affect the conclusions of this study.

We look forward to hearing from you regarding our submission. If you have further questions or suggestions, please do not hesitate to contact us.

---

## [Decision Letter · Decision Letter 1]

14 Dec 2020

Post-Intensive Care Syndrome as a Predictor of Mortality in Patients with Critical Illness: a cohort study

PONE-D-20-29849R1

Dear Dr. Kamiya,

We’re pleased to inform you that your manuscript has been judged scientifically suitable for publication and will be formally accepted for publication once it meets all outstanding technical requirements.

Kind regards,

Academic Editor

PLOS ONE

Additional Editor Comments (optional):

Reviewers' comments:

Reviewer's Responses to Questions

**Comments to the Author**

1. If the authors have adequately addressed your comments raised in a previous round of review and you feel that this manuscript is now acceptable for publication, you may indicate that here to bypass the “Comments to the Author” section, enter your conflict of interest statement in the “Confidential to Editor” section, and submit your "Accept" recommendation.

Reviewer #1: All comments have been addressed

Reviewer #2: All comments have been addressed

2. Is the manuscript technically sound, and do the data support the conclusions?

Reviewer #1: Yes

Reviewer #2: Yes

3. Has the statistical analysis been performed appropriately and rigorously? 

Reviewer #1: Yes

Reviewer #2: Yes

4. Have the authors made all data underlying the findings in their manuscript fully available?

Reviewer #1: Yes

Reviewer #2: Yes

5. Is the manuscript presented in an intelligible fashion and written in standard English?

Reviewer #1: Yes

Reviewer #2: Yes

6. Review Comments to the Author

Reviewer #1: I am satisfied with the answers to my comments, and have no further comment on the revised manuscript. Thank you.

Reviewer #2: The authors response well, so I have no more suggestion. I would recommend that the paper could be accepted in the present form.

7. PLOS authors have the option to publish the peer review history of their article (what does this mean?). If published, this will include your full peer review and any attached files.

Reviewer #1: **Yes: **Hsiu-Nien Shen

Reviewer #2: No